# Ozonated Sunflower Oil Exerted Potent Anti-Inflammatory Activities with Enhanced Wound Healing and Tissue Regeneration Abilities against Acute Toxicity of Carboxymethyllysine in Zebrafish with Improved Blood Lipid Profile

**DOI:** 10.3390/antiox12081625

**Published:** 2023-08-17

**Authors:** Kyung-Hyun Cho, Ji-Eun Kim, Ashutosh Bahuguna, Dae-Jin Kang

**Affiliations:** Raydel Research Institute, Medical Innovation Complex, Daegu 41061, Republic of Korea; ths01035@raydel.co.kr (J.-E.K.); ashubahuguna@raydel.co.kr (A.B.); daejin@raydel.co.kr (D.-J.K.)

**Keywords:** antioxidant, anti-inflammation, carboxymethyllysine, high-density lipoproteins, wound healing, tissue regenration

## Abstract

Ozonated sunflower oil (OSO) is an established therapeutic agent and nutraceutical harboring various therapeutic values, including antiallergic, derma-protective, and broad-spectrum antimicrobial activity. Conversely, the medicinal aspects of OSO for wound healing, tissue regeneration, and treatment of inflammation in dyslipidemia have yet to be fully elucidated. Herein, a comparative effect of OSO and sunflower oil (SO) was investigated to heal cutaneous wound and tissue regeneration of zebrafish impediment by carboxymethyllysine (CML) toxicity, following impact on hepatic inflammation and blood lipid profile. After OSO (final 2%, 1 μL) and SO’s (final 2%, 1 μL) treatment, substantial healing was elicited by OSO in the cutaneous wound of zebrafish impaired by CML (final 25 μg). As an important event of wound healing, OSO scavenges the reactive oxygen species (ROS), rescues the wound from oxidative injury, and triggers the essential molecular events for the wound closer. Furthermore, the intraperitoneal injection of OSO was noted to counter the CML-induced adversity and prompt tissue regeneration in the amputated tail fin of zebrafish. Additionally, OSO counters the CML-induced neurotoxicity and rescues the zebrafish from acute mortality and paralysis, along with meticulous recovery of hepatic inflammation, fatty liver changes, and diminished ROS and proinflammatory interleukin (IL)-6 production. Moreover, OSO efficiently ameliorated CML-induced dyslipidemia by alleviating the total blood cholesterol (TC), triglyceride (TG), and increasing high-density lipoproteins cholesterol (HDL-C). The outcome of multivariate assessment employing principal component analysis and hierarchical cluster analysis supports a superior therapeutic potential of OSO over SO against the clinical manifestation of CML. Conclusively, OSO owing to its antioxidant and anti-inflammatory potential, counters CML-induced toxicity and promotes wound healing, tissue regeneration, hepatoprotection, improved blood lipid profile, and survivability of zebrafish.

## 1. Introduction

Wound healing is a highly orchestrated process comprising four perpetual and concurrent phases [1]. Firstly, homeostasis is established, followed by an immediate inflammatory response with a marked accumulation of cytokine-producing leukocytes and antimicrobial function. Subsequently, the damaged area is cleaned, followed by wound compression as the remodeling of the extracellular matrix [1]. Mostly, the process of wound healing progressed normally; however, under certain physiological conditions, such as chronic disease, malnutrition, diabetes, and aging, this process is impaired, leading to chronic wounds and has a significant onus to patients and the medical system [1,2]. Severity can be assumed from the fact that approximately 4.5 million people in the USA are affected by chronic wounds [3]. Despite the disparity in etiology, chronic wounds shared certain common traits at the molecular level, comprising protraction and augmented inflammation characterized by abundant neutrophil infiltration, enhanced level of cytokines, proteases, reactive oxygen species (ROS), senescent cells, and enduring infection [3]. Chronic and non-healing wounds, such as foot ulcers, are the secondary complications associated with diabetes that inflict constant medication; even in many cases, amputation is required [4]. Diabetes is well characterized by the accumulation of advanced glycation end products (AGEs) such as carboxymethyllysine (CML) that integrate with skin collagen and alter epithelial cell migration and differentiation, consequently delaying wound healing [4]. In this context, different cutting-edge treatments comprise the application of extracellular matrices, engineered skin, negative pressure, and growth factors and have been broadly applied to treat chronic wounds [3]. 

Notably, ozone, owing to its antioxidant, antimicrobial, and immunomodulatory effect, has been projected as a potential agent for wound healing [5]. For instance, a marked impact of ozone therapy to accelerate wound healing and reduce amputation has been described in diabetic foot ulcers [6]. Interestingly, ozone therapy has been documented to reduce oxidative stress and inflammation, which are the established hallmarks of chronic wounds and, thus, provides a curative potential against chronic wounds [7]. It has been reported that ozone at 5–60 mg/L concentration is recommended for medical implementations where it exerts various therapeutic effects without having any deleterious ramifications [8,9]. In low doses, ozone stimulates cell protective activity [10], ameliorates tissue oxygenation, and activates the nuclear factor erythroid 2-related factor 2 (Nrf-2) pathway that regulates the expression of superoxide dismutase (SOD), catalase (CAT), reduced glutathione (GSH), and other endogenous antioxidants [10,11]. Amidst recent clinical studies demonstrated the persuasion of ozone on the phosphorylation of Nrf-2 and the activation of endogenous antioxidants that rescued multiple sclerosis patients from oxidative stress [12].

In support, numerous in vitro investigations have probed the influence of ozone on fibroblast migration and the manifestation of collagen-1, α-smooth muscle actin (α-SMA), epithelial-mesenchymal transition (EMT), and transforming growth factor β (TGF-β), an important biomarker for tissue repair, which strengths ozone applicability to deal with chronic wounds [13]. Moreover, ozone therapy is also documented for its potential to improve lipid profile, precisely by enhancing HDL-C and reducing LDL-C and TG [14]. Also, few studies have demonstrated the correlation of HDL with chronic wound healing via switching M1 to M2 macrophages and consequent downregulation of inflammatory markers such as iNOS, IL-6, and TNF [15,16]. Despite the several medicinal benefits, the short half-life of ozone is always a point of contention. It has been known that the half-life of ozone is only 40 min at 20 °C [11]. Fortunately, this can be meticulously improved up to 2 to 3 years by chemical entrapment of ozone in vegetable oils [17]. In vegetable oil, ozone is entrapped in the unsaturated site of fatty acids through the Criegee mechanism, yielding distinct ozonated compounds such as peroxides and ozonides [18,19]. For example, the successful entanglement of ozone in olive and sunflower oil is available commercially, displaying several medicinal worth’s, primarily linked to antimicrobial and antioxidant nature [20].

Of note, the substantial wound-healing activity of gaseous ozone [21], ozonated water [22], and ozone entrapped in olive oil [10] and sesame oil [23] have been described; however, the wound healing and tissue regenerative activity of ozonated sunflower oil (OSO) has not been studied extensively. Hitherto, most of the studies pertaining to OSO have primarily concentrated on its antimicrobial and germicidal properties [20,24]. We have previously documented Raydel OSO’s antimicrobial, cellular antioxidant [25], and anti-inflammatory potential [26] in zebrafish fed with a high cholesterol diet (HCD).

In continuation to our previous studies, the present study aimed to evaluate the healing potential of Raydel OSO on cutaneous wound stimulated by CML-induced oxidative stress and inflammation in adult zebrafish. Additionally, the role of Raydel OSO is tested for improvement of tail fin regeneration after amputation, survivability, hepatic function protection, and blood lipid profile amelioration in the CML-injected zebrafish.

## 2. Materials and Methods

### 2.1. Materials

Raydel ozonated sunflower oil (OSO) (Raydel^®^ Bodyone, Flambo oil) was complementary provided by Rainbow and Nature Pty, Ltd. (Thornleigh, NSW, Australia), which harbors the characteristic Oleozon^®^, as described previously (783.4 mmol peroxide/kg, acidity 2.42 mg KOH/g, and viscosity 131.2 mPa.s) [27]. The sunflower oil (SO) was procured from a retail store located in Daegu, Republic of Korea. N-ε-carboxylmethyllysine (CAS-No 941689-36-7, Cat#14580-5g), 2-phenoxyethanol (SigmaP1126; St. Louis, MO, USA), oil red O (Cat#O0625), and dihydroethidium (DHE, 104821-25-2, Cat #37291) were obtained from Sigma-Aldrich (St. Louis, MO, USA). Unless otherwise stated, all other chemicals and reagents were of analytical grade and used as supplied.

### 2.2. Zebrafish Husbandry

Zebrafish were nurtured per the accredited framework of care and use of laboratory animals [28,29], following the procedure approved by the Animal Care Committee and the Use of Raydel Research Institute (approval code RRI-20-003). Zebrafish were maintained at 28 °C (water temperature) at 14 h light and 10 h dark photoperiod in an automated water circulation tank (Bioengineering Company; Daejeon, Republic of Korea) and feed with normal tetrabits flake (TetrabitGmbh D49304, Melle, Germany; 47.5% crude protein, 6.5% crude fat, 2.0% crude fiber, 10.5% crude ash, containing vitamin A (29,770 IU/kg), vitamin D3 (1860 IU/kg), vitamin E (200 mg/kg), and vitamin C (137 mg/kg)).

### 2.3. Cutaneous Wound Formation in Adult Zebrafish and OSO Treatments

The wound healing effect of SO and Raydel OSO was assessed in 16-week-aged adult zebrafish. For wound generation, zebrafish were anaesthetized by drenching in 0.1% of 2 phenoxyethanol, followed by removal of the surface scale. A cutaneous wound of 2 mm diameter was notched into the left flank directly anterior to the anal and dorsal fin using a sterilized biopsy punch (Kai Industries co., Ltd., Oyana, Japan). The wounded zebrafish (n = 60) were randomly distributed into four groups (n = 15, each group). The cutaneous wound of zebrafish in Group I was topically treated with 1 μL of phosphate buffered saline pH 7.4 (PBS) (control), while the cutaneous wound of zebrafish in Group II was topically treated with 1 μL of 25 mg/mL CML in PBS (final 25 μg). In Groups III and IV, the cutaneous wound of zebrafish was treated with 1 μL SO (final 2%) and 1 μL OSO (final 2%) together with CML (25 μg final), respectively. After 3 min post-treatment, zebrafish from distinct groups (I–IV) were transferred into their respective chambers and maintained at 28 °C.

### 2.4. Visual Observation and Wound Healing

Zebrafish were constantly monitored, and the wound area was measured at 0, 2, 4, 6, 24, 48, 72, 120, 168, and 264 h post-treatment using methylene blue staining as earlier described method [30]. In brief, at different time points, zebrafish were anaesthetized, and the wound was stained with methylene blue solution (0.1% *w*/*v*, final 2 μL) for 1 min, followed by three-time washing and subsequent visualization under a stereomicroscope (Motic SMZ 168; Hong Kong, China). The wound area (blue stained) was computed employing ImageJ software (version 1.53r, http://rsb.info.nih.gov/ij/ accessed and retrieved on 16 May 2022). Percentage wound healing was quantified by comparing the wound area (mm^2^) calculated at 2, 4, 6, 24, 48, 72, 120, 168, and 264 h post-treatment with the wound area at the beginning (0 h).

### 2.5. Histological Analysis during Wound Healing

Histology analysis concerning morphological changes and oxidative stress in the wounded site was examined during wound healing. At 72- and 168-h post-treatment, 3 zebrafish from each group were retrieved, and tissue from the wounded site was surgically removed and fixed in 10% formaldehyde, followed by alcohol dehydration and amalgamation in paraffin. A 5 μm thick tissue section was prepared and stained with hematoxylin and eosin (H&E) [31] to visualize the morphological changes.

The oxidative stress status of the tissue was determined by the visualization of total reactive species (ROS) stained with dihydroethidium (DHE) as a previously described method [32]. The stained images were observed under fluorescent microscopy (Nikon Eclipse TE2000, Tyko, Japan) at 588 nm and 605 nm excitation and emission wavelength, respectively. 

### 2.6. Regeneration of Tail Fin

The comparative tissue regenerative effect of SO and Raydel OSO was evaluated in 16-week-aged adult zebrafish as previously described method [33]. In brief, zebrafish (n = 60) were anaesthetized by inundating in 0.1% of 2 phenoxyethanol, and the tail fin was amputated by a scalpel adjacent to dermal rays in the tail fin. Tail-fin amputated zebrafish were randomly segregated into 4 groups (n = 15, each group). The zebrafish in Group I and II received intraperitoneal injection of 10 μL PBS and 250 μg CML suspended in 10 μL PBS (analogue to 3 mM CML, contemplating ~zebrafish body weight 300 mg), respectively. Zebrafish in Group II and IV received 10 μL intraperitonial injection of CML (250 μg)+SO (final 2%) and CML (250 μg)+OSO (final 2%), respectively. The tail fin regeneration was examined under the stereomicroscope (Motic SMZ 168; Hong Kong) equipped with Motic cam2300 CCD camera (Motic cam2300 CCD). Images of the tail section were captured on successive days until day 7, and the tail fin regeared area was computed employing ImageJ software (version 1.53r, http://rsb.info.nih.gov/ij/ retrieved on 16 June 2022). 

### 2.7. Acute Inflammation in Zebrafish

Acute inflammation in adult zebrafish was instigated by injecting CML (250 μg/10 μL of PBS), as reported earlier [34]. Sixteen weeks aged adult zebrafish were randomly divided into 5 groups. Zebrafish in Groups I and II received intraperitoneal injections of 10 μL PBS and 10 μL CML (250 μg in PBS), respectively. Group II and III zebrafish received intraperitoneal injections of 250 μg CML suspended in 10 μL SO (final 1%) and SO (final 2%). Correspondingly, Group III and IV zebrafish were microinjected with 250 μg CML suspended individually in 10 μL of Raydel OSO (final 1% and 2%), respectively. All the groups received microinjections at the abdominal region using a 28-gauge needle after anesthetization by plunging in 0.1% 2-phenoxyethanol. The zebrafish survivability and swimming behavior were assessed at 30- and 60-min post-injection following the guidelines of OECD 2019 [35]. 

Zebrafish in various groups were sacrificed after 60-min post-injection using the hypothermic shock as described previously [26]; blood was collected immediately by the heart puncture from each zebrafish and combined with 3 μL of phosphate-buffered saline (PBS)-ethylenediaminetetraacetic acid (EDTA, final concentration, 1 mM), then collected in EDTA-treated tubes. The liver tissues were collected and preserved at ultra-low temperature (−70 °C) for histological investigations.

### 2.8. Plasma Analysis for Lipid Profile and Hepatic Function Biomarkers

The total cholesterol (TC) and triglyceride (TG) levels in the plasma were determined using a commercial assay kit (cholesterol, T-CHO, and TGs, Cleantech TS-S; Wako Pure Chemical, Osaka, Japan). HDL-C (AM-202), aspartate transaminase (AST), and alanine transaminase (ALT) were measured using a commercially available assay kit using AM-202, AM-103K, and AM102-K (Asan Pharmaceutical, Hwasung, Republic of Korea), correspondingly.

### 2.9. Histological and Immunohistochemical Investigation

The hepatic tissue from zebrafish was excised surgically and processed, as mentioned in Section 2.6. The hepatic morphological changes and ROS production were examined by H&E and DHE staining, respectively, as mentioned in Section 2.6. 

The pro-inflammatory IL-6 level in the hepatic tissue was determined by immunohistochemical (IHC) staining as previously described method [36]. In brief, the tissue section was flooded with 200× diluted IHC primary antibody (ab9324, Abcam, London, UK), followed by overnight incubation at 4 °C. The EnVision+ System-HRP polymer kit containing secondary antibody (1:1000, Code K4001, Dako, Denmark) developed the IHC stained area that was visualized under an ocular microscope (Nikon, Tokyo, Japan).

### 2.10. Statistical Analysis

All the experiments were accomplished in triplicates, and results are depicted as mean ± standard deviation. The statistical difference between the groups was determined using SPSS software (version 23.0; Statistical Package for the Social Sciences software program, Inc., Chicago, IL, USA), employing one-way analysis of variance (ANOVA) following post hoc examination (*p* < 0.05) using Tukey’s multiple range test. The multivariate exploratory techniques of principal component analyses (PCA) and hierarchical clustering (HCA) were performed using Minitab statistical software version 21.4.

## 3. Results

### 3.1. Raydel OSO Promoted Cutaneous Wound Healing and Suppress CML-Induced Oxidative Stress in Zebrafish

The SO and Raydel OSO-mediated wound healing effects were monitored in zebrafish at different time intervals, as represented in Figure 1. In PBS alone and CML+OSO2% treated zebrafish, the wound healing initiated at 4 h post-treatment, apparent by the reduced methylene blue stained area (Figure 1A). The utmost wound healing was noticed for the PBS-treated group no presence of CML, where the wound started to heal just after 4 h (18.9% wound closer) and progressively increased to 41.9% at 24 h, followed by 100% wound closer at 264 h post-treatment. Conversely, a significantly (*p* < 0.05) delayed wound healing was noticed in only CML-treated zebrafish, where no wound closure was noticed up to 24 h post-treatment, which slowly improved with time and attained 46.1% and 82.6% wound closer at 72 and 264 h post-treatment. Likewise, a similar wound healing pattern was noticed in the CML+SO2% treated zebrafish. In contrast, significantly (*p* < 0.05) better wound healing was observed in the CML+OSO2% treated zebrafish, where the wound started to close at 4 h post-treatment (10.9% wound closer), reached 39.6% at 6 h post-treatment, and further attained all almost 100% wound closer 264 h post-treatment. Noticeably, up to 6 h post-treatment, the wound healing in CML+OSO2% and PBS groups differed, afterwards; both the groups displayed a similar effect testifying to the excellent wound healing efficacy of OSO2% against the CML-induced stress.

Microscopically, the disappearance of the wound margin and the regeneration of pigments was first observed at 168 h post-treatment in the PBS-treated zebrafish which progressively intensified with time at 264 h (as indicated by red arrow) (Figure 1A). Contrarily, the disappearance of wound margin (as indicated by the blue arrow) and pigment regeneration was delayed in only CML-treated zebrafish, which was slightly improved by the treatment of SO2%. However, Raydel OSO2% displayed better restoration of pigments (as indicated by the red arrow) than only CML and CML+SO2% treated zebrafish, manifesting an expeditious scar-free wound healing potential of Raydel OSO2%.

The histology of the wounded tissue (skin and muscle) examined by H&E staining suggested a massive neutrophil infiltration in only CML-treated zebrafish (indicated by a dotted box, Figure 2A). Compared to this, a diminished neutrophil infiltration was observed in PBS-treated and CML+SO2% or OSO2% groups. Surprisingly, nonsignificant changes in epidermis thickness (neo-epithelization) were observed among all the groups during 72 h and 168 h post-treatment. However, a non-fragmented and regular formation of the epidermis was observed in the PBS-treated group. At the same time, a fragmented epidermis was observed in only the CML-treated group, which was effectively recovered by the treatment of OSO2% (Figure 2A). The granulation tissue formation (indicated by the black arrow) appeared first at 48 h, peaked at 72 h in the PBS-treated group, and then remained static or reduced to the basal level with further progression over time. Whereas the only CML-treated group displayed a higher granulation tissue formation that progressively intensified with time, suggesting the impaired wound healing process. Unlike this, diminished granulation tissue formation was observed in CML+SO2% or OSO2% treated groups during 72 h and 168 h post-treatment, suggesting better and prompt healing compared to only CML injected group. Furthermore, compact, muscular tissue (indicated by the blue arrow) was perceived in the PBS-injected group against a loosely arranged muscular tissue in only CML-treated group that was substantially recovered by the treatment of OSO2% at 72 h post-treatment (Figure 2A).

The DHE fluorescent staining demonstrated ROS production in the wounded site (Figure 2B,D). A non-significant difference in ROS generation was observed in the wounded tissue obtained from PBS, CML alone, and CML+SO2% groups at 72 h post-treatment. In contrast, a significant (*p* < 0.05) 3.5-, 4.4-, and 3.6-fold reduction in ROS level was noticed in the wounded tissue treated with OSO2% compared to PBS, CML alone, and CML+SO2% groups at 72 h post-treatment attesting substantial ROS inhibition property of OSO2%. Compared to 72 h, the DHE stained area corresponding to ROS production reduced from 3.7% to 1.4% in PBS treated group at 168 h post-treatment. Unlike this, at 168 h post-treatment, an enhanced ROS production was observed in CML alone; CML+ SO2% or OSO2% treated groups correspond to their ROS level at 72 h post-treatment. As opposed to only CML-treated group, a significantly (*p* < 0.05) 1.7-fold and 2.4-fold reduced ROS production was perceived in the CML+SO2% or OSO2% treated groups, respectively, at 168 h post-treatment. Noticeably, a significant (*p* < 0.05) 1.4-fold reduced ROS production was observed in the wounded tissue treated with OSO2% compared to SO2%, demonstrating the supremacy of OSO2% to inhibit ROS generation and, consequently, wound healing.

### 3.2. Raydel OSO Ameliorated Tail Fin Regeneration in CML-Injected Zebrafish

The tissue regenerative role of SO and Raydel OSO against CML stimulated challenge in the amputated tail fin is depicted in Figure 3. As compared to the PBS group (5.6 ± 0.51 mm^2^), CML injected group (2.8 ± 0.65 mm^2^) displayed a significant (*p* < 0.05) 2-fold reduced tail fin regeneration at 3 days post-treatment, signifying the impact of CML to delayed tail fin regeneration. The treatment of OSO2% efficiently counters CML-induced adversity, evident by a significantly (*p* < 0.05) 1.4- and 1.6-fold better tail fin regeneration than only the CML injected group at 4 and 5 days post-injection, respectively. Surprisingly, no modulatory effect of SO2% against CML-impaired tail fin regeneration was observed. The findings strongly imply that OSO2% substantially affects CML-impaired tissue regeneration.

### 3.3. Raydel OSO Rescued Zebrafish against CML-Evoked Fatality and Acute Paralysis

Severe mortality with merely 50% survivability was observed in only CML injected group, suggesting the high lethality of CML (Figure 4A). Treatment of SO1% and SO2% also had no impact on the survivability of zebrafish against the toxicity posed by CML. In contrast, treatment of OSO1% and OSO2% prevents the CML-incited mortality evident by 1.4-fold and 1.9-fold higher zebrafish survivability, in comparison to only CML injected group (Figure 4A). While compared with the SO2% group, the survivability of zebrafish was 1.4 times better in the OSO2% group at 60 min post-injection, signifying the impact of ozonolysis on the functionality of SO.

As depicted in Figure 4B, severe paralysis was observed in only CML injected group, where nearly all the zebrafish swimming activity ceased at 30 min post-injection. Likewise, SO1%, SO2%, and OSO1% treatment had no significant impact on the restoration of zebrafish swimming activity hampered by CML injection (Figure 4B,C). In contrast, a significant (*p* < 0.05) recovery with 43.3% restoration in swimming behavior was observed in OSO2% injected zebrafish at 30 min post-injection that further elevated to 96.6% at 60 min post-injection exhibiting 5.4-fold and 3.6-fold rejuvenation than only CML injected group. Compared to SO2%, a significantly (*p* < 0.05) 2.6-fold better restoration of swing activity was observed in the OSO2% group at 60 min post-injection. These findings suggested the preemptive role of OSO2% against CML-induced paralysis that subsequently enhanced the survivability of zebrafish.

### 3.4. Raydel OSO Suppressed Inflammation, ROS Production and Restore Hepatic Function Biomarkers in CML-Injected Zebrafish

As exemplified in Figure 5A, a visible hepatic degeneration and massive neutrophil infiltration around the bile duct (indicated by black arrow) and arterial area (indicated by blue arrow) was perceived in the CML injected group that accounts for 36.2% H&E-stained area. The injection of SO1%, SO2%, and OSO1% efficiently prevents the CML-induced hepatic degeneration and neutrophil infiltration apparent by 28.7%, 30.1%, and 29.4% H&E-stained area, respectively, which is approximately 18% lower than the H&E-stained area of only CML injected group. Nonetheless, CML co-injected with OSO2% displayed the most profound effect, as evidenced by the least neutrophil infiltration around the portal vein (marked by a red arrow). The 18.6% H&E-stained area, which is significantly 49.1% (*p =* 0.001) lower than the stained area that appeared in the CML injected group, signifies the hepatoprotective effect of OSO2%. While compared with SO2%, 39.2% (*p =* 0.001) reduced H&E-stained area was detected in response to OSO2%, suggesting the higher hepatoprotective efficacy of OSO2%. 

As depicted in Figure 5B, severe fatty liver changes indicated by 15.2% oil red O-stained area was noticed in the CML-treated group, which was substantially prevented by SO2%, OSO1%, and OSO2% treatments. A significantly 22.2% (*p =* 0.034) reduced oil red O-stained area was quantified in the OSO2% injected group compared to only CML injected group. However, the most promising effect on the restoration of CML altered fatty liver changes displayed by treatment of OSO1% and OSO2% manifest by 33.1% (*p =* 0.004) and 46.9% (*p* < 0.001) lower oil red O-stained area than the only CML injected group. While comparison to SO2%, a significant 31.8% (*p =* 0.027) reduced oil red O-stained area in response to OSO2%, signifying the superior potency of OSO2% to prevent CML-stimulated fatty liver changes. 

The DHE fluorescent staining was employed to evaluate the ROS production (Figure 5C). The utmost ROS production with 48.45% DHE stained area was perceived in only CML injected group. Treatment of SO1% and SO2% displayed a non-significant effect against CML-induced ROS production. Contrary to this, OSO1% and OSO2% significantly diminished CML-induced ROS production. Compared to the only CML-injected group, a significantly 1.6-fold alleviated ROS production was observed in the OSO1%-treated group. In contrast, a more profound effect with a significantly 6.8-fold (*p* < 0.001) and 5.3-fold (*p* < 0.001) lower ROS production was observed in the OSO2% treated group against only CML and CML+SO2% treated groups, respectively, establishing the exalted strength of OSO2% to counter CML induced ROS generation.

Figure 5D illustrates the IHC stained area corresponding to IL-6 generation in various groups. A 16.5% IHC stained area was spotted in only CML injected group, which was non-significantly affected by the treatment of SO1%, SO2%, and OSO1%. In contrast, OSO2% substantially alleviated the CML-induced IL-6 production apparent by a diminished IHC stained area (11.7%) that is significantly 1.4-fold (*p =* 0.02) and 1.5-fold (*p* < 0.03) lower than the IHC stained area for only CML and CML+SO2% injected groups, asserting the impact of OSO2% on IL-6 production.

The plasma analysis suggested a significant (*p* < 0.05) elevation of hepatic injury biomarkers (AST and ALT) in response to CML, which was restored substantially using SO and OSO injection (Figure 6A). A significant (*p* < 0.05) 28.1% and 21.9% reduced AST level than the only CML injected group was observed in SO1% and SO2% injected groups. Consistent with this, OSO1% and OSO2% injection significantly (*p* < 0.05) 33.1% and 26.6% alleviated AST levels against only CML injected group. Contrary to AST, a profound impact with 63.1%, 59.1%, 64.3%, and 63.4% reduced serum ALT levels were assessed in SO1%, SO2%, OSO1%, and OSO2% injected groups, respectively, concerning only CML injected group (Figure 6B). 

A combined result of hepatic function biomarkers and histological analysis attested that OSO, mainly OSO2%, restored the CML-impaired hepatic damage and equilibrated IL-6 and ROS generation.

### 3.5. Raydel OSO Improved the CML-Induced Dyslipidemia

As depicted in Figure 7, the minimum level of HDL-C was observed in only the CML injected group, which was slightly enhanced in response to SO1% and SO2% treatment. Juxtapose, a significantly (*p* < 0.05) 106.2% and 38.8% higher HDL-C was detected in OSO1% and OSO2% treated groups against only CML injected group. Surprisingly, OSO1% showed the most promising effect, with significantly (*p* < 0.05) 32.8% elevated HDL-C compared to OSO2% treatment. Unlike HDL-C, a non-significant effect of SO2%, OSO1%, and OSO2% injection was noticed for TC level against only CML injected group. Contrary to this, significantly (*p* < 0.05) 14.7%, 99.9%, and 49.5% higher ratios of HDL-C/TC were noticed amid SO2%, OSO1%, and OSO2% injected groups, respectively, than the only CML injected group. Interestingly, compared to OSO2%, a 24.8% higher HDL-C/TC level was noticed in response to SOS1%. Further, as compared to only CML injected group, significantly (*p* < 0.05) 20.1%, 26.7%, and 34.9% reduced TG levels were detected in SO2%, OSO1%, and OSO2% injected groups, respectively. Nonetheless, no impact of SO1% was observed to reduce the CML-induced TC level. Also, the CML-induced higher TG/HDL-C level was significantly (*p* < 0.05) reduced by 17.4% and 32% by the injection of SO1% and SO2%. Nevertheless, the most profound effect was demonstrated by OSO1% and OSO2%, which accounted for 64.5% and 52.9% reduced TG/HDL-C levels than the CML injected group. In addition, a substantial effect of OSO1% and OSO2% was observed in reducing non-HDL-C levels, apparent by a significantly (*p* < 0.05) 59.9% and 35.7% reduced non-HDL-C levels, respectively, against only CML injected group. Conversely, a non-significant effect of SO1% and SO2% treatment on alleviating CML-induced non-HDL-C levels was observed. The results implied OSO’s competency to synchronize the CML altered plasma lipid profile of zebrafish.

### 3.6. Multivariate Analysis

Multivariate analysis concerning principal component analysis (PCA) and hierarchical cluster analysis (HCA) was employed to segregate the effect of SO1%, SO2%, OSO1%, and OSO2% on zebrafish survivability, swimming activity, hepatic markers, and plasma lipid profile altered by the CML treatment. PCA results revealed that PC1 and PC2 cover 86% of the variance and segregated the effect of CML+OSO1% and CML+OSO2% with only CML and CML+SO1% or SO2% injected groups (Figure 8A,B). The negative PC-1 and PC-2 values only for OSO2% suggested that OSO2% has a distinct impact against CML-induced hepatoxicity and altered lipid profile compared to other groups that eventually improved zebrafish survivability and swimming activity.

Furthermore, the HCA was performed, which is regarded as a powerful statistical tool to arrange the data based on the similarity between the groups. Based on the outcome of the different tested hepatic parameters, plasma lipid profile, swimming behavior, and survivability of zebrafish, the HCA dendrogram separated all the groups into two major clusters (Figure 8C). The CML+SO1% and SO2% were grouped in the same cluster closely associated with only CML injected group, signifying their effect is somewhat close to the effect of only the CML injected group. Contrary to this, CML+OSO1% and OSO2% groups are clustered differently and are distant from the other groups signifying their distinct impact against the CML-induced toxicity.

## 4. Discussion

Wounds, especially chronic wounds, are a global problem that puts significant health and economic burden [37]. There are different advancements in dealing with chronic wounds. However, most of them have certain limitations owing to the complexity of chronic wounds, where multiple events participate that varies from wound to wound. However, extravagant oxidative stress and enduring inflammation are the worthiest attributes among all the chronic wounds that provoke assorted molecular events, leading to delayed wound healing [38]. Therefore, a curative agent must harbor pleiotropic pursuit with momentous antioxidant and anti-inflammatory activity. Sunflower oil (SO) is a rich source of diverse phytoconstituents, tocopherols, vitamin E, niacin, folic acid, and minerals (calcium, copper, iron, magnesium, selenium, sodium, and potassium) with notable functionality, including derma protective and antimicrobial activity [39]. Fortunately, the proficiency of SO can be ameliorated prodigiously by ozonation. 

Accumulative literature has cataloged OSO’s diverse functionality, including antioxidant and antimicrobial activity [40,41]. Recently, we have accentuated the antioxidant role of Raydel OSO that thwarts macrophages from apoptotic cell death [25] and elevates zebrafish survivability, dyslipidemia, and hepatic steatosis altered by high cholesterol diet [26]. Despite the several perks, OSO has not been probed meticulously for wound healing and tissue regeneration employing zebrafish, that considered the exemplary model for preclinical investigations [42,43]. Nonetheless, few preliminary observations on turtles substantiated the wound-healing role of OSO [44]. Pertaining to this, we delved into the comparative wound healing, tissue regenerative, and hepatoprotective efficiency of SO and OSO against CML-posed adversity in zebrafish. CML is a well-known oxidative stressor [38] associated with diabetic and cardiovascular diseases and often ties with a collagen matrix, leading to compromised wound healing [4]. Herein, the CML was used to instigate oxidative stress and inflammation (to emulate the chronic wound environment), and the therapeutic implication of SO and Raydel OSO was investigated.

The collected results suggested that zebrafish treated with Raydel OSO emerge with rapid cutaneous wound healing impaired by the exposure of CML. Nevertheless, compared to SO, a profound wound-healing effect by OSO, implying the modulatory impact of ozonation on the activity of SO. The apparent reason for the difference in the healing activity is the presence of ozonized and other ozone-catalyzed compounds in OSO that are well-known for varied biological activity [19,20]. The results align with the previous reports implying the impact of ozonation on the ameliorated functionality of SO [25,26]. The better wound healing activity of Raydel OSO is strengthened by the earlier report deciphering the impact of ozone therapy on diabetic foot ulcers by the cellular induction of vascular endothelial growth factor (VEGF), platelets-derived growth factors (PDGF), and transforming growth factor-β (TGF-β) [13]. Following similar effects, we believe entrapped ozone in OSO leads to prompt wound recovery against the adversity posed by the CML. The wound-healing activity of tocopherols [45] and folic acid [46] has been well delineated. The presence of tocopherols and folic acid in SO [39] may synergistically work with ozone-catalyzed compounds in OSO and be a supplementary reason for the superior wound-healing activity of OSO.

The H&E and DHE staining of the wounded tissue documented massive neutrophil infiltration and ROS production in response to CML that SO and OSO proficiently neutralized. Moreover, a time-dependent augmentation of ROS in response to CML was observed that designates high oxidative stress, a hallmark of chronic wounds. Treatment of SO and OSO, precisely OSO, efficiently thwarts the CML-induced ROS inclining toward wound healing with time. The results of DHE staining clearly imply that CML-induced oxidative stress is the main culprit behind delayed wound healing, and the antioxidant property of OSO is a key event that leads to expedited wound recovery. The findings collectively indicated that OSO, owing to its vigorous antioxidant character, counters the CML-impelled oxidative stress and provides a lucrative environment for wound healing. Results comply with our preceding reports documenting OSO’s free radical scavenging potential and cellular antioxidant activity [25,26]. Additionally, several studies have cataloged the impact of ozone on the induction of nuclear factor erythroid 2-related factor 2 (Nrf2) that coordinates the synthesis of cellular antioxidants to counter oxidative stress [47,48]. 

Furthermore, it has been well established that around 60% of chronic wounds face bacterial colonization at the wounded site [49]. Therefore, a substance with a broad range of antimicrobial action in addition to antioxidant and tissue regenerative activity is more coveted. Fortunately, OSO has an intense antibacterial activity [25], invigorating OSO prospects as therapeutic to cure chronic wounds. 

Besides the cutaneous wound healing, a comparative role of SO and Raydel OSO on tail fin regeneration and CML-induced acute paralysis was determined. Like the cutaneous wound healing, OSO2% prompted the amputated tail fin regeneration impaired by the CML injection. Contrary to this, no effect of SO2% on tail fin regeneration was observed, manifesting the influence of ozonation on the functionality of SO. The rigorous mechanism behind the tissue-productive potential of ozonated oil has yet to be fully understood. Presumably, the OSO’s antioxidant [25] and anti-inflammatory [26] properties lead to curative events for tissue generation. The notion is in accordance with an earlier report documenting the inflammation regulatory effect of ozonated water expedites the caudal fin regeneration of zebrafish [50].

Furthermore, OSO2% displayed a remarkable resilience against CML-induced acute paralysis, evidenced by rapid restoration of swimming activity. The precise mechanism behind the improved swimming activity and survivability has yet to be discovered. Nevertheless, OSO’s anti-inflammatory and antioxidant properties are crucial to rescuing zebrafish from CML-induced toxicity. The cognition is strongly backed by our earlier finding describing the inhibition of proinflammatory IL-6 by CIGB-258, and tocilizumab leads to the recovery of zebrafish from acute paralysis [51]. Additionally, OSO2% displayed hepatoprotection against CML-induced toxicity, apparent by reduced neutrophil infiltration, as observed in H&E staining. Also, the oil red O staining demonstrates the preclusion of fatty liver changes by OSO2%. Likewise, OSO2% efficiently curtailed CML-induced IL-6 and ROS generation, indicating OSO’s anti-inflammatory and antioxidant role (Figure 9).

Proinflammatory IL-6 is a pivotal cytokine allied with inflammatory diseases [52] and acute paralysis [51]. The low hepatic IL-6 level in response to OSO2% justifies that IL-6 inhibition by Raydel OSO is the primary reason for rescuing zebrafish against CML-induced acute paralysis and mortality. Evermore, several studies acknowledged a positive correlation between IL-6, liver fibrosis [53], and fatty liver changes [54]. Herein, we have also observed a similar trend that inhibiting IL-6 by OSO improves CML-impaired fatty liver changes. The antioxidant nature of Raydel OSO may be a reason for the lower IL-6 production, supported by the literature describing oxidative stress as a provocative factor for the induction of inflammatory pathways [55]. 

The Raydel OSO efficiently curtailed the TG, TC, and non-HDL-C and enhanced the HDL-C level, thus impact against CML-induced dyslipidemia. We assumed that diminished IL-6 production in response to OSO is the key event that balances the serum lipid profile. The notion following preceding reports suggests an association of IL-6 with an altered lipid profile [56,57]. In inflammatory diseases such as rheumatoid arthritis (RA), elevated IL-6 levels are allied with decreased HDL-C and HDL-associated protein apoA-1 [56,58]. Similarly, augmented TC and alleviated levels of HDL were witnessed in psoriasis and systemic lupus erythematosus (SLE) [57]. Many curative drugs for psoriasis, SLE, and RA balance the blood lipid profile, signifying a correlation between inflammation and serum lipid profile [57]. Herein, it was perceived that Raydel OSO inhibits the IL-6 production, which consequently amends dyslipidemia, precisely elevating HDL level.

Elevated HDL can be correlated with wound healing, as many recent reports established an association between HDL and chronic wound healing via modulation of inflammation [59]. Perpetually, one clinical study also defines a correlation between endogenous HDL and diabetic wound healing [60]. In addition, tropical application of HDL was found competent to cure diabetic and non-diabetic wounds [61,62]. These reports strengthen the current findings that Raydel OSO’s impact on HDL can prompt wound healing. More interestingly, in our previous study, we documented the influence of Raydel OSO on the structural alteration of HDL_3_ [25], which leads to the amelioration of paraoxonase-1 functionality, an important biomarker to balance oxidative stress and inflammation [63], strengthening the notion that Raydel OSO, due to HDL_3_ structural alteration and elevating HDL level, stimulates chronic wound healing.

## 5. Conclusions

Raydel OSO exhibited a substantial healing and tissue regenerative role against severely impaired wounds by CML in zebrafish. While compared to SO, the substantially higher therapeutic efficacy of Raydel OSO concludes ozonation is a vital tool to ameliorate the functionality of SO. Raydel OSO’s antioxidant and anti-inflammatory role is logged as a key event behind the prompt wound repair and the therapeutic effect against CML-induced mortality and paralysis, hepatotoxicity, and dyslipidemia. Additionally, the noteworthy persuade of Raydel OSO on ameliorating HDL-C levels plausibly be associated with wound healing, which needs to be addressed in detail. The multi-faceted role of Raydel OSO substantiates its viability as a therapeutic agent for chronic wound healing, countering oxidative stress, inflammation, and the associated consequences.

## Figures and Tables

**Figure 1 antioxidants-12-01625-f001:**
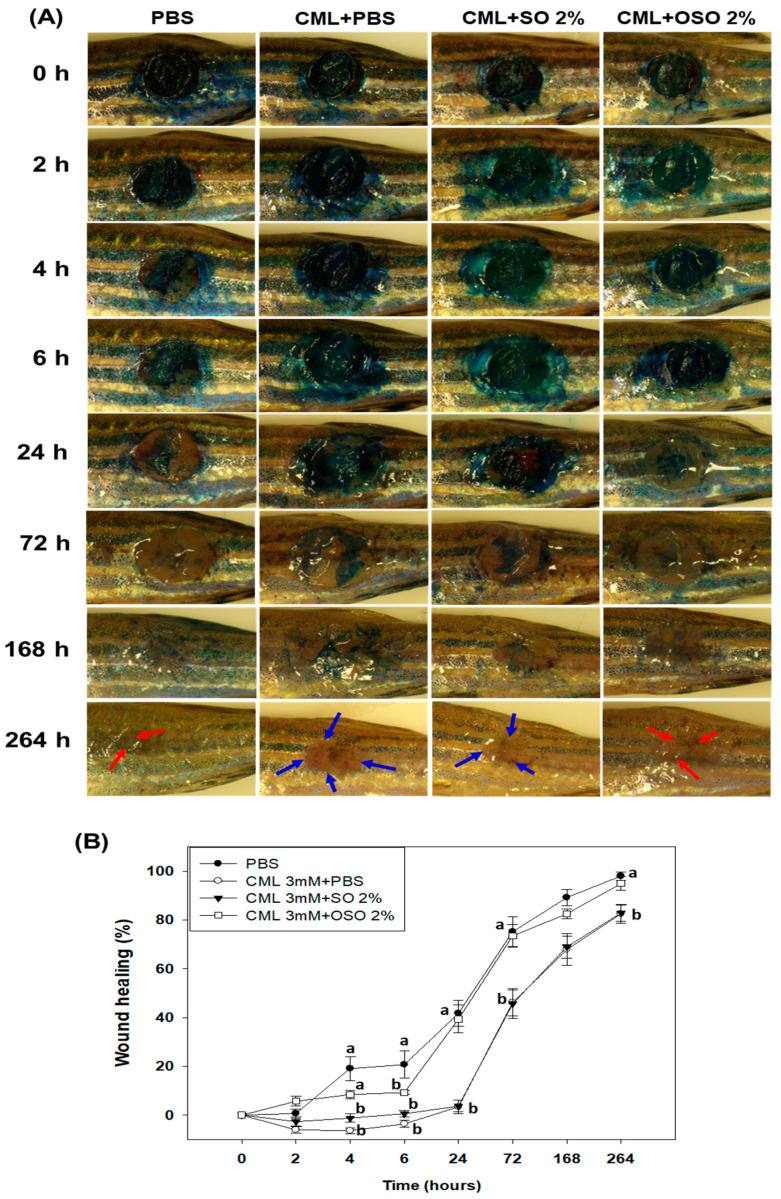
The comparative healing effect of sunflower oil (SO) and Raydel ozonated sunflower oil (OSO) against carboxymethyllysine (CML) exposed cutaneous wound in adult zebrafish. (**A**) Pictorial representation of wound stained with methylene blue (0.1% *w*/*v*) and pigment formation during 264 h post-treatment with 6.7× magnifications. The red arrow implies the pigment formation in the wounded area, while the blue arrow represents the wound scar. (**B**) Percentage wound closer during 264 h post-treatment. The percentage of the wound closer was computed by comparing the stained wound area measured at different times with respect to the wound area at 0 h. The PBS control group received a topical dose of PBS only, CML+PBS groups received a topical dose of CML (25 μg) dissolved in PBS, while the CML+SO2% and CML+OSO2% groups were co-treated with CML (25 μg) with SO2% and OSO2%, respectively. The letters (a,b) above the graphs indicate the statistical difference (*p* < 0.05) among the groups at the given time.

**Figure 2 antioxidants-12-01625-f002:**
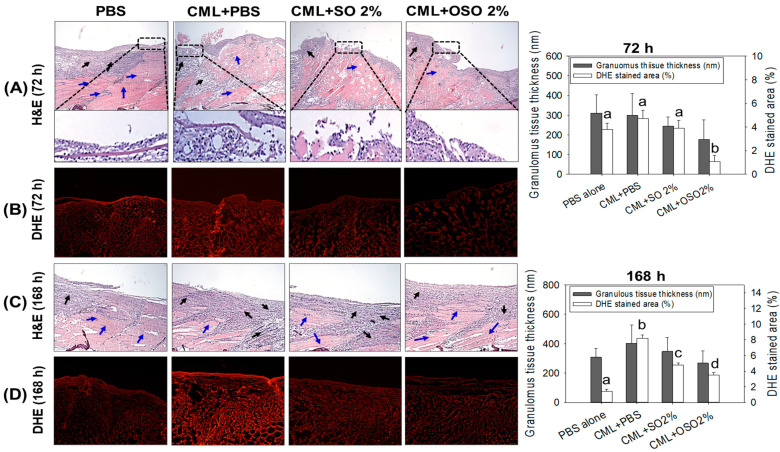
Comparative effect of sunflower oil (SO) and Raydel ozonated sunflower oil (OSO) on skin morphology and reactive oxygen species (ROS) production on carboxymethyllysine (CML) treated cutaneous wound in adult zebrafish. (**A**,**C**) Portray hematoxylin and eosin (H&E) staining at 72 h- and 168 h post-treatment with 400× magnification The black and blue arrows represent the granulation and muscular tissue, respectively. (**B**,**D**) Portray dihydroethidium (DHE) staining for ROS production at 72 h and 168 h post-treatment 400× magnification. The PBS (control) group revied a topical application of PBS only, CML+PBS groups received a topical dose of CML (25 μg) dissolved in PBS, while the CML+SO2% and CML+OSO2% groups were co-treated with CML (25 μg) with SO2% and OSO2%, respectively. The letters (a–d) above the graphs indicate a statistical difference (*p* < 0.05) among the groups at the given time.

**Figure 3 antioxidants-12-01625-f003:**
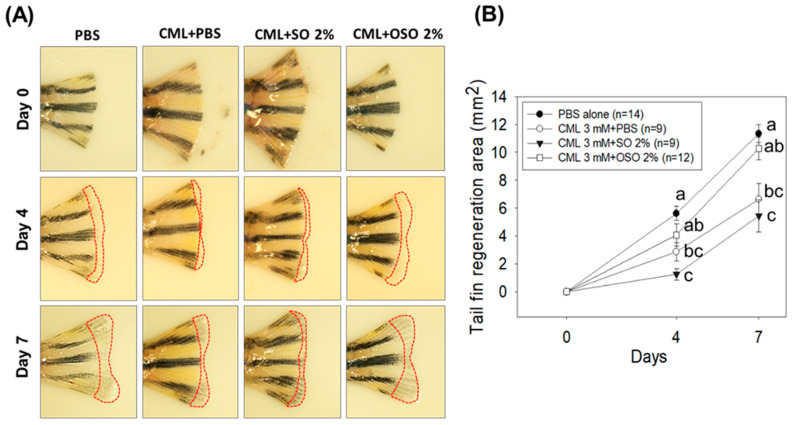
A comparative tail fin regenerative effect of sunflower oil (SO) and Raydel ozonated sunflower oil (OSO) in carboxymethyllysine (CML) injected adult zebrafish. (**A**) Morphology of tail fin amid 7 days post-treatment. The red dotted line indicates the regenerated tissue at the proximal end. (**B**) kinetics of tail fin regenerated area. The PBS group was microinjected with PBS (vehicle), and the CML+PBS group was microinjected with CML (3 mM) dissolved in PBS. CML+SO2% and CML+OSO2% groups were microinjected with CML (3 mM) together with SO2% and OSO2%, respectively. The letters (a–c) above the graphs indicate the statistical difference (*p* < 0.05) among the groups at the given time.

**Figure 4 antioxidants-12-01625-f004:**
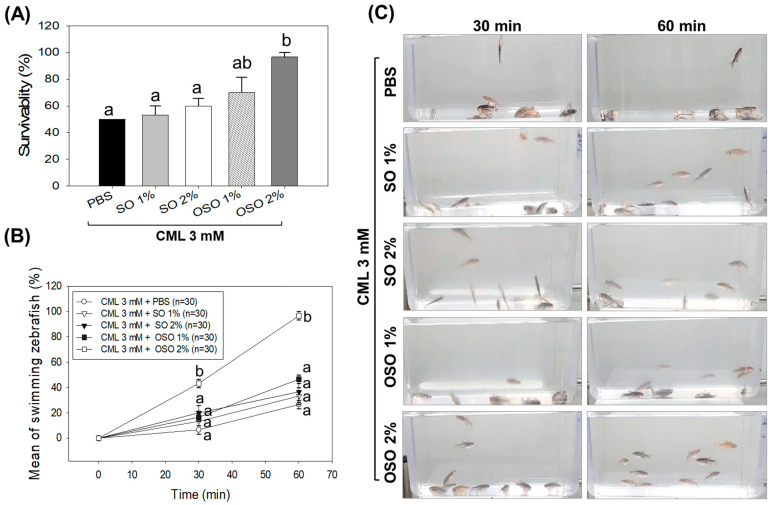
A comparative swimming behavior and survivability of zebrafish injected with carboxymethyllysine (CML) and subsequently treated with sunflower oil (SO) and Raydel ozonated sunflower oil (OSO). (**A**) Zebrafish survivability at 60 min post-injection. (**B**) Mean value of percentage swimming activity and (**C**) snapshots of swimming activity at 30 min and 60 min post-injection. The CML+PBS group was injected with CML (250 μg) dissolved in PBS, whereas CML+(SO1%, SO2%, OSO1%, and OSO2%) groups were co-injected with CML (250 μg) along with either SO1% or SO2% or OSO1% or OSO2%. The letters (a,b) above the graphs indicate the statistical difference (*p* < 0.05) among the groups at the given time.

**Figure 5 antioxidants-12-01625-f005:**
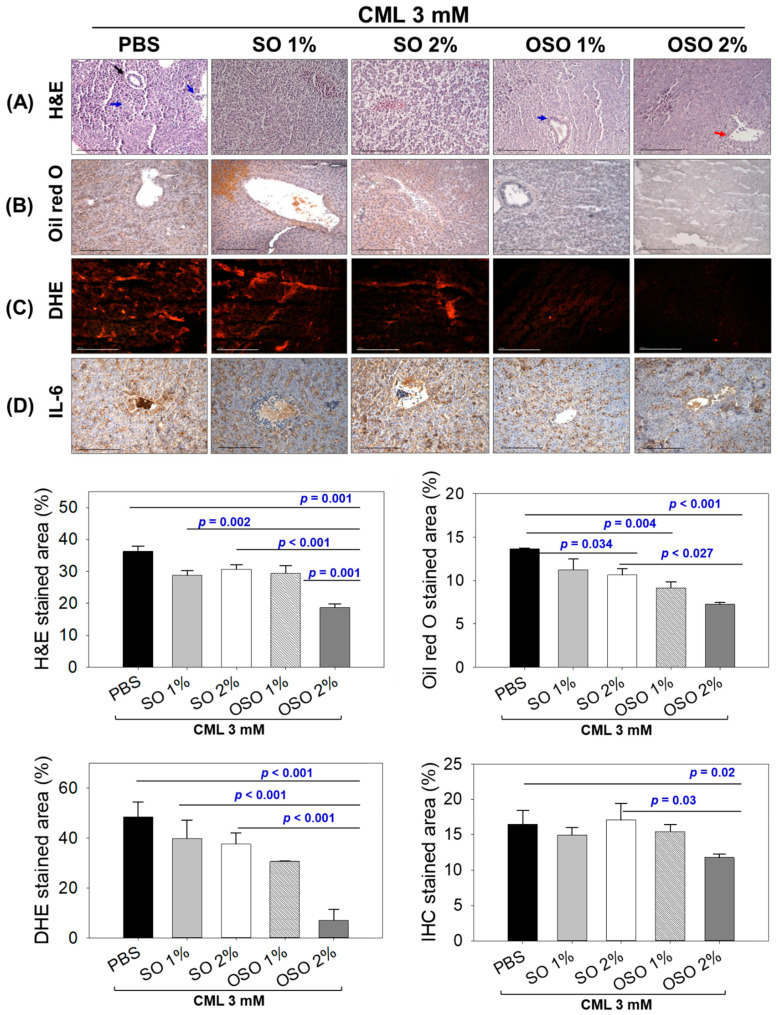
Histology of hepatic tissue deciphering a comparative effect of sunflower oil (SO) and Raydel ozonated sunflower oil (OSO) on hepatic morphology, fatty liver changes, oxidative stress, and IL-6 production in carboxymethyllysine (CML) injected adult zebrafish at 60 min post-injection. (**A**) Hematoxylin and eosin (H&E) staining. The black, blue, and red arrows represent the bile duct, arterial area, and portal vein. (**B**) Oil red O staining. (**C**) DHE staining for reactive oxygen species detection. (**D**) IL-6 production [detected by immunocytochemistry (IHC)]. The depicted images are 400× magnified [graphic scale = 0.1 mm], and the stained area was computed employing ImageJ software (version 1.53, http://rsb.info.nih.gov/ij/ retrieved on 16 May 2022). The CML+PBS group was injected with CML (3 mM) dissolved in PBS, whereas CML+(SO1%, SO2%, OSO1%, and OSO2%) groups were co-injected with CML (3 mM) along with either SO1% or SO2% or OSO1% or OSO2%. *p* value documented the pairwise statistical variation retrieved from the ANOVA employing the Turkey’s test for post hoc analysis.

**Figure 6 antioxidants-12-01625-f006:**
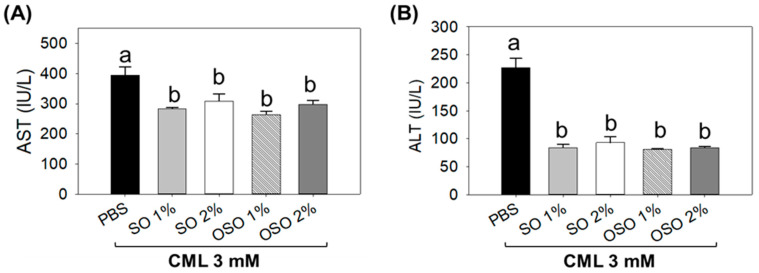
Effect of sunflower oil (SO) and Raydel ozonated sunflower oil (OSO) on the liver function biomarkers (**A**) aspartate aminotransferase (AST) and (**B**) alanine aminotransferase (ALT) in carboxymethyllysine (CML) injected adult zebrafish. The CML+PBS group was injected with CML (3 mM) dissolved in PBS, whereas CML+(SO1%, SO2%, OSO1%, and OSO2%) groups were co-injected with CML (3 mM) along with either SO1% or SO2% or OSO1% or OSO2%. The letters (a,b) above the graphs indicate the statistical difference (*p* < 0.05) among the groups at the given time.

**Figure 7 antioxidants-12-01625-f007:**
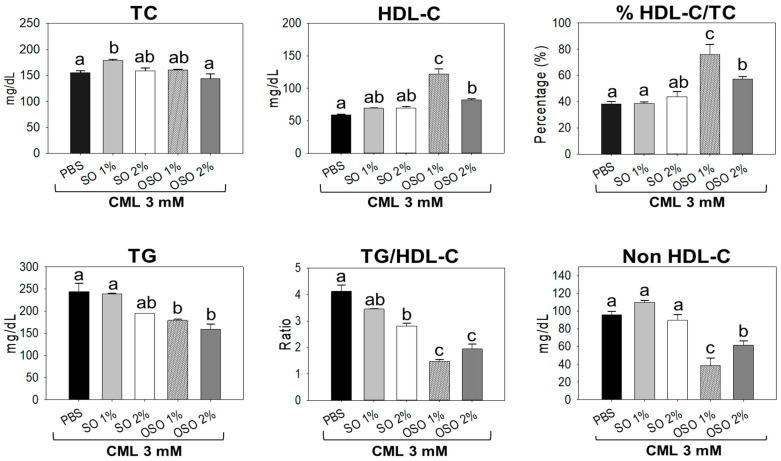
Comparative analysis of sunflower oil (SO) and Raydel ozonated sunflower oil (OSO) on blood lipid profile of the carboxymethyllysine (CML) injected adult zebrafish. The CML+PBS group was injected with CML (3 mM) dissolved in PBS, whereas CML+(SO1%, SO2%, OSO1%, and OSO2%) groups were co-injected with CML (3 mM) along with either SO1% or SO2% or OSO1% or OSO2%. The letters (a–c) above the graphs indicate the statistical difference (*p* < 0.05) among the groups at the given time. The TC, HDL-C, and TG are acronyms for total cholesterol, triglyceride, and high-density lipoprotein cholesterol.

**Figure 8 antioxidants-12-01625-f008:**
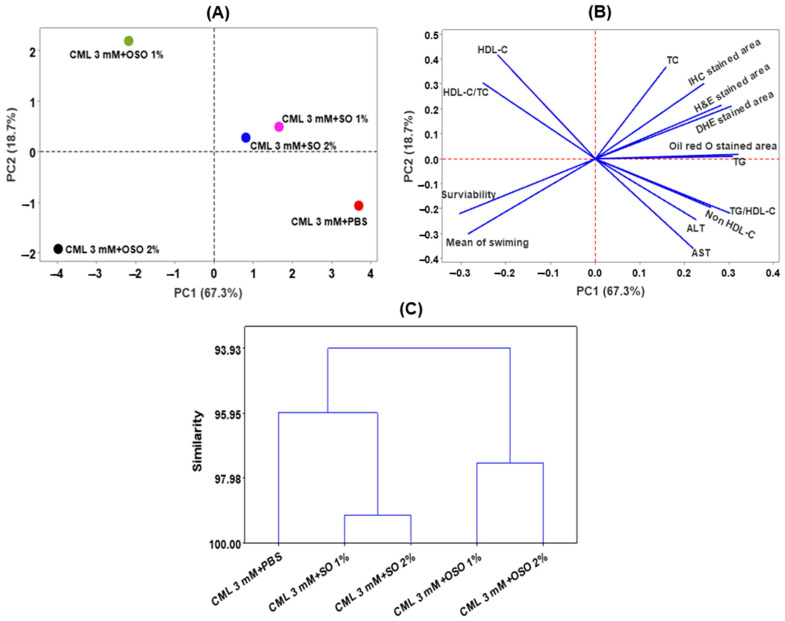
Multivariate analysis based on tested parameters (survivability, swimming, liver function biomarkers, histology, and blood lipid profile) of zebrafish stimulated by carboxymethyllysine (CML) and subsequently treated with sunflower oil (SO) and Raydel ozonated sunflower oil (OSO). (**A**,**B**) are the scoring plot and loading plot, respectively, obtained from principal component analysis (PCA). (**C**) Hierarchical cluster analysis (HCA). The PCA and HCA were performed using Minitab statistical software version 21.4. The CML+PBS group was injected with CML (3 mM) dissolved in PBS, whereas CML+(SO1%, SO2%, OSO1%, and OSO2%) groups were co-injected with CML (3 mM) and SO1% or SO2% or OSO1% or OSO2%. AST, aspartate aminotransferase; ALT, alanine aminotransferase; HDL-C, high-density lipoprotein cholesterol; TC, total cholesterol; TG, triglyceride.

**Figure 9 antioxidants-12-01625-f009:**
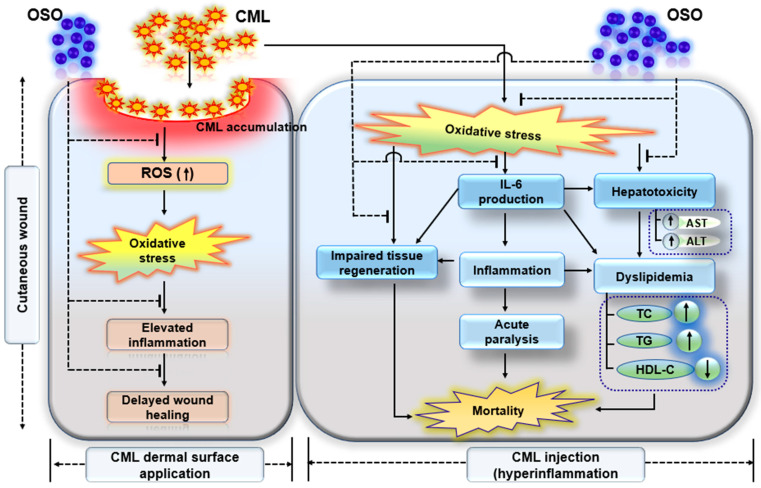
Summary of Raydel ozonated sunflower (OSO) mediated events that counters carboxymethyllysine (CML) impelled hazards leads to cutaneous wound healing, tissue regeneration and hepatoprotection in zebrafish.

## Data Availability

The data used to support the findings of this study are available from the corresponding author upon reasonable request.

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
