# Peer review of "Ozonated Sunflower Oil Exerted Potent Anti-Inflammatory Activities with Enhanced Wound Healing and Tissue Regeneration Abilities against Acute Toxicity of Carboxymethyllysine in Zebrafish with Improved Blood Lipid Profile"

_antioxidants, 2023, doi:10.3390/antiox12081625_

Round 1
Reviewer 1 Report
The paper could be accepted after major revision. Authors mentioned that ozone has antioxidant properties. However, ozone is a reactive oxygen species (ROS) that can interact with cellular components and cause oxidative stress. Thus, authors should discuss (add references in Introduction/Discussion) how ozone could exert antioxidant functions. Maybe it could be mediated by controlled toxicity produced by low concentrations of O3, which enhance the cell's suppliance of antioxidant properties without causing any further damage etc.
What is therapeutic concentration range of ozone and what is ozone’ concentration in the ozonated sunflower oil? Also, authors mentioned that half-life of ozone could be increased by unsaturated substances.
What is half-life of ozone in the sunflower oil?
Minor comments:
Line 70: instead TNF-α should be TNF (without alpha). Because LTα is no longer referred to as TNFβ, TNFα, as the previous gene symbol, is now simply called TNF, as shown in HGNC (HUGO Gene Nomenclature Committee) database.
Line 105: printed “...Korea). with”
Line 178: abbreviation for PBS should be at first appearance in the text.
Line 430: use capital letters for the title in the section 3.5.
Author Response
Thank you for your valuable comments and suggestions.
Please find attached doc as point-to-point response.

Reviewer 2 Report
In this manuscript, the authors demonstrated that ozonated sunflower oil exhibited a substantial healing and tissue regenerative role against wounds induced by carboxymethyllysine in zebrafish. The data clearly showed that the therapeutic effects of ozonated sunflower oil were apparently better than those of sunflower oil in several bioassays. The manuscript provides encouraging information, and thus is recommended for publication. Some suggestion listed below for the consideration of revision.
1. Ozone therapy has been documented to reduce oxidative stress and inflammation, which are the established hallmarks of chronic wounds and, thus, provides a curative potential against chronic wounds [References 5-7 in the manuscript]. In addition, ozonated sunflower oil, but not sunflower oil, was shown to possess therapeutic effects against carboxymethyllysine- induced wounds in zebrafish in this manuscript. Unfortunately, ozone alone was not examined in this study. The authors have to explain why ozonated sunflower oil instead of ozone alone was used in this study.
2. Lines 503-505: Sunflower oil (SO) is a rich source of diverse phytoconstituents and minerals with notable functionality, including derma protective and antimicrobial activity [32]. The authors are suggested to describe major or functional phytoconstituents and minerals of sunflower oil, possibly related to the therapeutic effects against carboxymethyllysine- induced wounds in zebrafish in this manuscript. Were these therapeutic effects activated or enhanced by ozonation? Alternatively, was ozone therapy enhanced by phytoconstituents and minerals of sunflower oil?
3. Line 77: As indicated by the authors, ozone has been entrapped in olive oil [15] and sesame oil [16] for wound-healing. Did the authors compare the effects of sunflower oil with those of olive oil or sesame oil? Are the authors sure that sunflower oil is superior to olive oil and sesame oil?
4. Lines 272-275: “Furthermore, compact, muscular ……. at 72 h post-treatment (Figure 2 A).” Only one sentence was stated in this paragraph. It might be incorporated into the previous paragraph.
Author Response

(The authors gave the same response as above.)

Round 2
Reviewer 1 Report
The paper could be accepted.